# WaveletGPT: Wavelet Inspired LLMs

## Abstract

Large Language Models (LLMs) have ushered in a new wave of artificial intelligence advancements impacting every scientific field and discipline. We live in a world where the data around us, e.g., text, audio, and music, has a multi-scale structure associated with it. This paper infuses LLMs with a traditional signal processing idea, wavelets, during pre-training to take advantage of the structure. Without adding **any extra parameters** to a GPT-style LLM architecture in academic setup, we achieve the same pre-training performance almost twice as fast in text, raw audio, and symbolic music. This is achieved by imposing a structure on intermediate embeddings. When trained for the same number of training steps, we achieve significant gains in performance, comparable to pre-training a larger neural architecture. Our architecture allows every next token prediction, access to intermediate embeddings at different temporal resolutions in every layer. This work will hopefully pave the way for incorporating multi-rate signal processing ideas into traditional LLM pre-training. Further, we showcase pushing model performance by improving internal structure instead of just going after scale.

## 1 Introduction and Related Work

LLMs have ushered in a super-renaissance of AI advancements and are touching every scientific and engineering discipline. At the heart of this is the Transformer architecture (Vaswani et al., 2017), initially proposed for machine translation. Transformer architecture became the backbone of GPT (Generative Pretrained Transformer) language models (Brown et, 2020) first proposed by Open-AI. Modern LLMs are trained on a straightforward objective: To predict the next token given the previous context, preserving the causality. This not only works for language but also for robotics (Brohan et al., 2023b;a), protein sequences (Madani et al., 2020), raw audio waveforms(Verma & Chafe, 2021), acoustic and music tokens (Huang et al., 2019; Verma & Smith, 2020; Borsos et al., 2023), videos (Yan et al., 2021) etc. This simple recipe of tokenization/creating an embedding and feeding it to transformers also has given rise to non-causal architectures such as BERT(Devlin et al., 2019), Vision Transformers (Dosovitskiy et al., 2021), Audio Transformers (Verma & Berger, 2021) and Video Transformers (Selva et al., 2023). The recent surge in multi-modal LLMs similar to Gemini family (Team et al., 2023) or Chameleon Chameleon (2024) would pave the way computers able to reason like humans. With increased performance by scale, LLMs are reaching hundreds of billions of parameters (Brown et, 2020) with Google's Switch Transformer even reaching trillion parameters (Fedus et al., 2022). Recent concerns suggest AI research is shifting from academia to industry, according to a Washington Post article by Nix (2024). The theme of this work is to enhance LLM capabilities to match those of larger architectures or achieve equivalent performance in fewer training steps. We extract intermediate embeddings from each decoder block and impose a hierarchical multi-scale structure without adding parameters. The signals across tokens in the intermediate layers are extracted, which we modify, similar to wavelet decomposition, while maintaining causality (Figure 1). Unlike previous techniques that enhance smaller models with larger ones, our approach focuses on improving performance during pre-training. A common approach is knowledge distillation Hinton et al. (2015), where a larger model guides a smaller one. Gu et al. (2024) used KL divergence to enhance next-token prediction from teacher model feedback, still relying on a powerful model rather than training the smaller one from scratch. A line of work, such as Nawrot et al. (2022), proposed hierarchical transformers using upsampling-downsampling operations, similar to the hourglass U-Net architecture Long et al. (2015) in computer vision. This approach achieves comparable results to Transformers but with more efficient computation. Clockwork RNN (Koutnik et al., 2014) improves long-context modeling by splitting RNN neurons into modules that update at different clock rates.

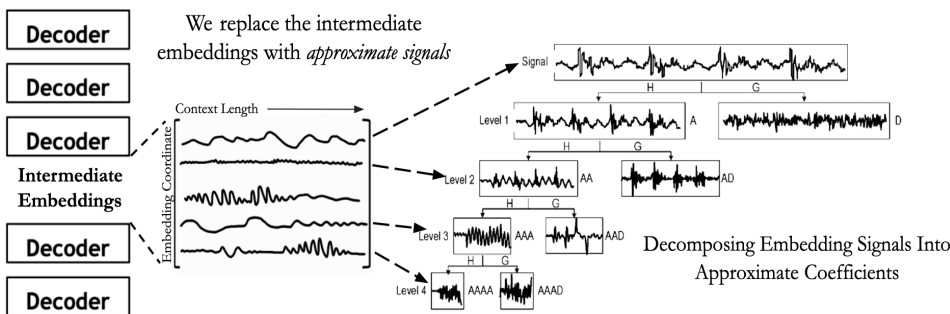

Figure 1: Manipulating signals in between decoder blocks of GPT. For each of these signals we compute a 1-D causal discrete haar wavelet transform or learnable approximation at different levels to mimic the multi-scale structure that exists for text, raw audio, symbolic music. Fig on right is from Gao & Yan (2006), which gives a detailed account of non-stationary signal processing for 1-D signals. We take the leftmost route of approximate coefficients, which allows us coarsest to finest scales.

Only a few modules activate at each time step, enabling efficient learning of long-term dependencies. In contrast, our approach modifies intermediate embeddings with simple tweaks, without using separate learning modules or varying update rates. Clockwork RNN (Koutnik et al., 2014) improves long-context modeling by splitting RNN neurons into modules that update at different clock rates. Only a few modules activate at each time step, enabling efficient learning of long-term dependencies. In contrast, our approach modifies intermediate embeddings with simple tweaks, without using separate learning modules or varying update rates. Model pruning (Sun et al., 2024) removes weights based on their salience, to match the same performance as a large model like LLAMA (Touvron et al., 2023), with fewer compute flops during inference. However, this approach still relies on starting with a pre-trained large model than training from scratch. We also exclude quantization methods like Dettmers et al. (2024), which focus on improving inference or fine-tuning existing models. The other line of work is tinkering with the intermediate embeddings. Tamkin et. al (2020) proposed hand-tuned filters on the Discrete Cosine Transform- DCT over the entire context length (Ahmed et al., 1974) of the latent space for different NLP tasks for non-causal BERT (Devlin et al., 2019), making them not applicable for causal applications such as language modeling. There have been work on applying ideas from signal processing-like methods to BERT-like non-causal architectures. We discuss two here, FNet and WavSPA. They focus on improving attention, which is different from our work on GPT which retains vanilla attention layer. FNet proposed by Lee-Thorp et al. (2022) removes the costly attention mechanism replacing with a 2-D FFT block. This operation is non-causal as it looks into future tokens for computing 2-D FFT. WavSpA (Zhuang et al., 2024) carries attention mechanism in the wavelet space. Since the wavelet transform is a multi-resolution, capturing long-term dependencies at various time scales, the input sequences are transformed into wavelet space, and the attention mechanism is carried out and then reconstructed. However, computing wavelet transform is non-causal, making them non applicable for GPT based LLMs as they look at the entire sequence length for capturing variations from coarsest to finest scales (as can be seen in Figure 1 of (Zhuang et al., 2024)). This paper modifies only the intermediate embeddings of a LLM model. Our work is inspired by neuroscience, which provides evidence that the human brain learns multi-scale representations for language at multiple time scales (Caucheteux et al., 2023) instead of fixed resolution representations. Our paper explicitly imposes multi-scale representation onto every intermediate decoder embedding at different dimensions. The contribution of the paper is as follows:

- We propose, to the best of our knowledge, the first instance of incorporating wavelets into LLM pretraining. We add multi-scale filters onto each of the intermediate embeddings of decoder layers using Haar/learnable wavelet pipeline. This allows every next token prediction access to multi-scale intermediate embeddings in every decoder layer instead of fixed-resolution representations.
- We show to speed the pre-training of a shrunk down GPT like transformer-based LLM in the range of 40-60%, with adding multi-scale structure. With the same number of training steps, the model gives a substantial non-trivial performance boost, akin to adding several layers or parameters.

## 2 DATASET

We use three open-source datasets from three different domains: natural language, symbolic music, and raw audio waveform. For text, we choose text-8 (Mikolov et al., 2012). We choose this over other datasets as i)it is popular and widely cited character-level language modeling dataset and ii) use a simple vocabulary (space + 26 lowercase characters) to detach the effects of various tokenizers. It has 100M characters with split training split as given by Al-Rfou et al. (2019). For raw audio, the goal is predicting the next sample given context. We use the YouTube-Mix-8 dataset, used for long-context modeling (Goel et al., 2022; Verma, 2022). Our vocabulary size is 256, with a sampling rate 16KHz as input is 8-bit. We use a third dataset, MAESTRO (Hawthorne et al., 2019), containing over 1000 MIDI files of classical music pieces. We use tokenizer proposed by Huang et al. (2019), which converts MIDI tracks into discrete tokens with a vocabulary size 388. The goal in all three modalities is not to chase state-of-the-art performance, as *this paper was written in an academic setting with very few computational resources*. We compare pre-training performance to the shrunk-down version of GPT with/without adding multi-scale structure to the embeddings using Haar or learnable kernels.

## 3 METHODOLOGY

This section will describe the approach to incorporating wavelets into transformer-based Large Language models while retaining the causality assumption. The ideas described here are generic and can be easily extrapolated to setups without a Transformer architecture e.g. state space architectures.

### 3.1 INCORPORATING WAVELETS INTO INTERMEDIATE EMBEDDINGS

For any signal, we compute a version of the discrete wavelet transform and incorporate it back into the signal. Let $x^l_{(i)}$ be the output of the $l^{th}$ decoder layer, representing the activation along the $i^{th}$ coordinate, with a dimension equal to the context length $L$ of the transformer-based GPT model. In the original GPT architecture with $N + 1$ layers and embedding dimension $E$, we obtain $N \cdot E$ signals of length $L$ from intermediate embeddings between decoder blocks, where $E$ ranges from $[0 - 128)$ dimensions. A wavelet is a signal with zero mean and non-zero norm, designed to address the limitations of traditional Fourier-based representation. For any signal $x[n]$, the discrete wavelet transform resembles passing the signal through filters of varying resolutions, as illustrated in Figure 2. We will use the Haar wavelet, a family of square-shaped functions, throughout this paper, obtained from a mother wavelet via scaling and shifting operations. Given a mother wavelet function $\psi$, we come up with the child wavelets as $\psi_{j,k}[n]$, where $j$ is the scaling factor and $k$ the shift factor.

$$\psi_{j,k}[n] = \frac{1}{\sqrt{2^j}}\psi\left(\frac{n - k2^j}{2^j}\right) \tag{1}$$

These signals are shifted and scaled to capture information at various time scales, with $n$ representing time or the context length. This concept resembles the diagram in Figure 1, which illustrates capturing different signals in the intermediate layers of Transformer decoders at various resolutions. We now define the discrete wavelet transform, which passes any signal through filters and downsampling operations. This process, shown in Figure 2, is similar to a convolutional neural network (CNN) like ResNet (He et. al, 2016), featuring learned convolutional filters analogous to $h[n]$ and $g[n]$, along with downsampling, such as max pooling. In traditional convolutional architectures, we typically follow one branch of Figure 2, recursively taking the output of filters and downsampling. This similarity contributed to the popularity of wavelets in the 1990s and 2000s for image understanding, reflecting parallels with convolutional architectures (Huang & Aviyente, 2008; Kingsbury & Magarey, 1998). As we use Haar wavelets, this involves passing the signal through low-pass and high-pass filters corresponding to the kernels $g[n]$ and $h[n]$. The Haar wavelet transform averages and computes differences, with impulse responses $g[n] = \left[\frac{1}{2}, \frac{1}{2}\right]$ and $h[n] = \left[\frac{1}{2}, -\frac{1}{2}\right]$. Figure 2 provides a detailed explanation of the discrete wavelet transform. For a 1-D signal $x[n]$ of length $L$, we obtain level 1 coefficients by applying filters $g[n]$ and $h[n]$, followed by downsampling. Thus, the approximation coefficients $y_{approx}$ and $y_{detail}$ result from an LTI system defined by convolution followed by downsampling by two, seen in Equation 2. This behavior is reflected in convolution in Algorithm 1.

$$y_{\text{approx}}[n] = \sum_{k=-\infty}^{\infty} x[k]g[2n - k] \quad ; \quad y_{\text{detail}}[n] = \sum_{k=-\infty}^{\infty} x[k]h[2n - k] \tag{2}$$

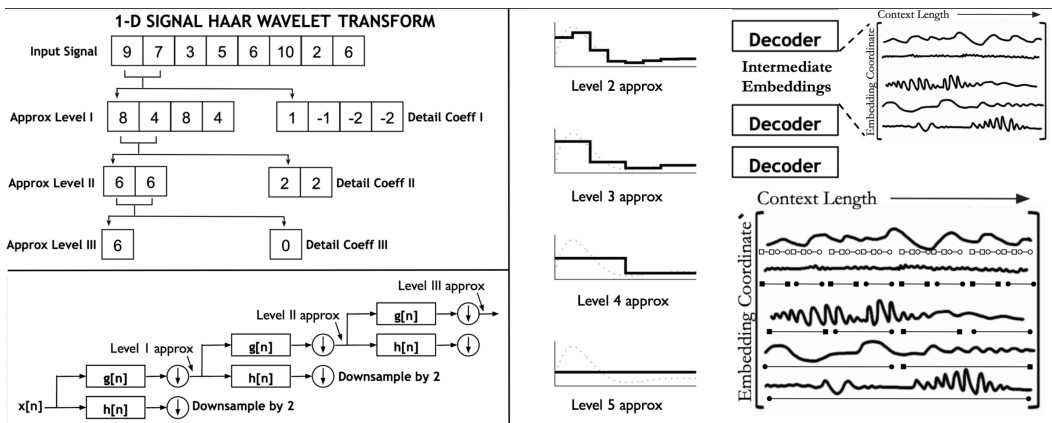

Figure 2: (Bottom L): A tree structure depicting 3-level filter bank that gives us a signal at different resolutions. We get approximate coefficients by passing it through an impulse response corresponding to the chosen wavelet and recursively down-sampling it. (Top left) Computing the approximate and detailed coefficients at various levels. We recursively take first-order averages/differences followed by downsampling until we get only a single scalar representative of the input signal. (Right) For a 32-length signal different levels of approximate coefficients of haar wavelet capture the signal from the coarsest to finest. The figure on left are redrawn from (Flores-Mangas, 2014). (R) We compute embeddings moving at different rates via causal wavelet approximation, where certain embedding dimensions evolve at the coarsest level (similar to level 5), while others follow a finer resolution (level 2). This infuses multi-scale information in all embeddings for decoder layers for every token.

To obtain multi-scale representations of the original signal, the operation for $x[n]$ is recursively applied to $y_a$ (approx) to derive level 2 wavelet coefficients $y_a^2$ and $y_d^2$ (detail). Here, $x[n]$ represents intermediate signals across the context length at each decoder block output in the LLM. The approximate coefficients $y_a$ and $y_d$, along with their decompositions $\{y_a, y_d, y_a^2, y_a^3, y_a^4, \ldots\}$, are used for further processing. Notably, $y_a^2, y_a^3, y_a^4$ have lengths reduced by factors of $2, 4, 8, \ldots$. The Haar wavelet transform averages adjacent samples while preserving causality by averaging current and past samples. Higher-order coefficients capture averages over larger context lengths, as shown in Figure 2. We can continue until only a single scalar value remains, representing the mean of the signal. The Haar wavelet transform computes averages and differences to create a multi-resolution representation, capturing low and high frequencies at different resolutions. Figure 2 illustrates the same signal captured at coarser and finer representations using Haar wavelets, applied to intermediate embeddings, allowing each next token prediction access to these representations. For the case of learnable wavelet kernels, we create a multi-resolution representation by varying the kernel size (Algorithm 1) to allow the LLM to learn the optimal kernels optimized for next token prediction.

## 3.2 CONNECTING WAVELETS AND LLM EMBEDDINGS

In many signal processing applications, first-order detail coefficients and approximate coefficients help understand signals at various levels. We aim to do the same but with signals from intermediate transformer embeddings across tokens. However, we focus only on approximate coefficients. Our premise is that real-world data is structured—text ranges from letters to words, sentences, and topics, while symbolic music ranges from notes to motifs and pieces. Using the Haar wavelet, this can be approximated as a simple averaging operation, as described earlier. For the learnable version, we allow weights of the kernel for multi-scale version to be optimized according to how best we can predict the next token. Continuing with the approximate coefficients will eventually yield a single scalar, the average of the entire signal in the case of the Haar wavelet. To match the original signal's sequence length from the approximation coefficients, several methods can be employed, including up-sampling. For clarity, we refer to the signal approximated at a specific level with the same length as the "approximate signal" at that level, distinguishing it from the shorter approximate coefficients. In Figure 2 (R), to obtain the signal approximation at various levels matching the original

input signal $x[n]$, we apply the wavelet kernel by multiplying the approximate coefficients with the kernel for that level (e.g., $[1, 1]$, $[1, 1, 1, 1]$, etc.). This is illustrated in the piecewise constant function shown in Figure 2. Different LLM embedding coordinates define unique resolution kernels, each corresponding to a specific scale for data capture. The reconstructed signal $x_{\text{recon}}[n]$, a method to derive the *approximate signal*, is computed from wavelet coefficients $c_j$ at level $j$ as:

$$x^j_{\text{recon}}[n] = \sum_k c_k \cdot \psi_{j,k}[n] \qquad (3)$$

Equation 3 requires storing child wavelets at various approximations, complicating the process and rendering it non-causal as computing $c_k$ takes into account the entire signal. Due to the dependence of $c_k$ to future information, we cannot use this to reconstruct the signal from its approximate coefficients. To adapt this for LLM we simplify the computation of the *approximate signal* in a differentiable manner using a variant of the equation from Equation 3 in both multi-resolution learnable/non-learnable kernels settings. For the Haar wavelet, we compute a average of the input signal with varying kernel lengths, increasing the length until it approximates the entire signal. The kernel length determines the level of signal approximation. LLMs operate under a causality assumption, modifying the signal at a location using prior samples within the kernel length. We zero-pad the signal to the left when window length is shorter than kernel. Wavelet transform at different levels gives several versions of the signal at different resolution which can mess up the structure of the intermediate Transformer embeddings. To address this, we create different resolutions for signal approximations parameterized by the embedding dimension. In Section 4.4, we make these kernels learnable, allowing the architecture to maintain multi-scale operation (Equation 3), with learnable weights with $x_{\text{recon}}[n]$ now being learned. The resolution is parameterized by the embedding coordinate is described next.

---

**Algorithm 1** Wavelet-GPT

$E$: Model or Embedding Dimension
$L$: Context Length
$N + 1$: Number of Decoder Layers
**for** layer $l = 1, 2, \ldots, N$ **do**
    $\mathbf{x}^l \leftarrow$ Output of Transformer $l^{th}$ Decoder Block              //Dimension $E$ x $L$
    $\mathbf{xn}^l \leftarrow$ Modified Transformer Embedding Replacing $\mathbf{x}^l$
    $\mathbf{xn}^l_{(i)} \leftarrow \mathbf{x}^l_{(i)}$    For Embedding dimension    $i > E/2$

    $\mathbf{f(i)} \leftarrow 2^F$ where //Finding kernel length function of embedding coordinate nearest power of 2
            $F = int(L_k * (i - E/2)/(E/2 - 1))$    $L_k = \lfloor \log_2(L) \rfloor + 1$    $i <= E/2$

    $\mathbf{xn}^l_{(i)}(\mathbf{k}) \leftarrow \frac{1}{\mathbf{f(i)}} \sum_{\mathbf{m=k-f(i)}}^{\mathbf{k}} \mathbf{x}^l_{(i)}(\mathbf{m})$    $i <= E/2$    // For Non-learnable fixed Haar wavelet
    $\mathbf{xn}^l_{(i)}(\mathbf{k}) \leftarrow \sum_{\mathbf{m=0}}^{\mathbf{f(i)-1}} \mathbf{h(m)} \cdot \mathbf{x}^l_{(i)}(\mathbf{k - m})$    $i <= E/2$ //For learnable wavelet kernel $h$
**end for**

---

### 3.3 Wavelet Coefficients by Embedding Dimension Coordinates

One option is to compute the *approximate signals* for each coordinate signal $x^l_{(i)}$ across all decoder layers at levels I to IX. For a context length of 512, this would require nine additional signals with resolutions of 512, 256, 128, 64, 32, 16, 8, 4, and 2, significantly increasing the architecture's complexity and necessitating major modifications to our GPT model. To address this, we propose a novel solution: instead of computing all levels of *approximate signals* for every intermediate embedding dimension, we parameterize the level by the embedding dimension index. We want to steer the embeddings only a little into the inductive biases we impose to avoid too much tinkering with that they learn. Transformers have been wildly successful without incorporating any inductive biases. Ideally, we want the best of both worlds, nudging intermediate GPT embeddings in only half of the dimensions. We adjust intermediate GPT embeddings in only half the dimensions. Embeddings from $E/2$ to $E$ (coordinates 64 to 128 when $E = 128$) remain unchanged. For the rest, we apply processing based on their index $i$. Mathematically, if $x^l(i)$ is an intermediate embedding after the $l^{th}$ decoder layer along the $i^{th}$ dimension, the modified signal $xn^l(i)$ equals $x^l_{(i)}$ for $i \in [E/2, E]$. For $0 \le i < E/2$, we impose structure using an approximate signal, calculated from wavelet coefficients corresponding to the index $i$. We use a mapping function $f$ that takes coordinate $i$ (ranging from

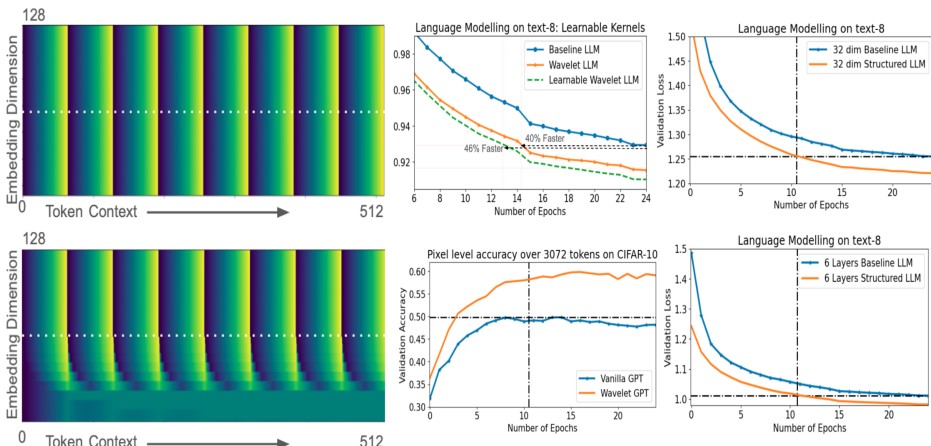

Figure 3: (Left) Toy example showing how variations in how embeddings move along token dimension, and how we impose multi-rate structure where different embedding dimensions advance at distinct rates while maintaining causality. Latent space now learns at varying rates for each token, with patterns dispersing from dimension 64 to 0. (Right) Validation loss during pre-training on text-8 with learnable multiscale structure. The model achieves comparable performance nearly twice as fast. When trained for the same number of epochs, we get a performance boost akin to adding additional decoder layers. We also demonstrate the architecture's performance on text-8 with a 32-dim model, matching the speedup similar seenf for a 128-dim model and for a shallower six layers. For the LRA image benchmark, we observe a 10% performance increase without adding extra parameters.

0 to $E/2$) and returns the kernel size corresponding to approximation levels from $I$ to $IX$. The linear function gradually increases from level $I$ (kernel size 2 at $i = 0$) to level $IX$ (kernel size 512 at $i = E/2$, or the coarsest representation for a generic case, i.e., a scalar). Now, let us find out how we compute the modified new signal $xn^l_{(i)}$ that replaces the original intermediate Transformer embeddings $x^l_{(i)}$. $f(i)$ denotes the kernel size for the coordinate $i$. Now, the modified signal is:

$$xn^l_{(i)} = x^l_{(i)} \text{ for } i > E/2 \quad ; \quad xn^l_{(i)}(k) = \frac{1}{f(i)} \sum_{m=k-f(i)}^{k} x^l_{(i)}(m). \tag{4}$$

For cases where $k - f(i) < 0$, we zero-pad the signal to ensure valid average/kernel computation. Specifically, for the Haar wavelet, the modified signal acts as a causal moving average filter with finite length, averaging the embedding signal along the $i^{\text{th}}$ coordinate with a kernel size determined by $f(i)$. This operation does not introduce new parameters, maintaining causality in LLMs and preventing future token leakage as seen in Equation 4. We can extend this approach to learn an optimal kernel specific to the task. In Algorithm 1, each value of the modified signal at token $k$ is computed using a convolution with a learned kernel $h(.)$ and variable length $f(i)$, parameterized by the embdding coordinate dimension $i$. Each kernel is learned independently for every signal in LLM.

### 3.4 IMPOSING STRUCTURE: TOY EXAMPLE

In Figure 3, we illustrate a toy example of how we impose structure onto decoder Transformer embeddings. The left side shows eight variations along the token dimension, with onset/sudden bursts at token indices 32, 64, etc., decreasing to zero before rising again. As discussed in the introduction, datasets inherently possess a hierarchical structure, which we capture by imposing it on intermediate Transformer embeddings at each layer. In this example, we retain embeddings at the original resolution for half the dimensions (split by a white line). For the other half, we gradually increase the kernel length across the context and compute the average causally. The final embedding dimension averages over the token dimension with a kernel size equal to the context length (zero-padding if necessary). This creates highways, allowing embeddings to move at different rates: the coordinates from $E/2$ to $E$ move at the Transformer's original speed, while those from 0 to

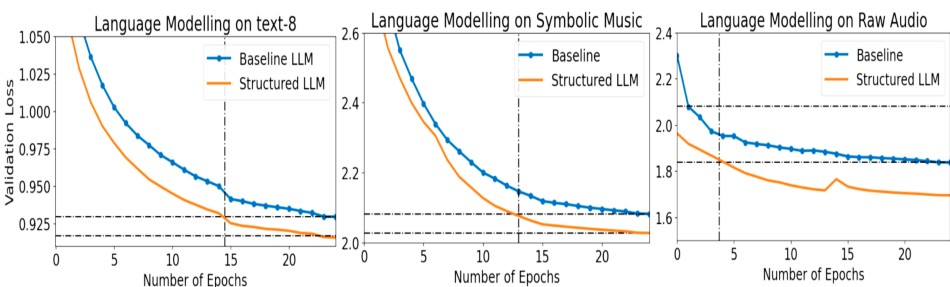

Figure 4: Results for three modalities: natural language, symbolic music, and raw audio. We see that we achieve much faster performance than the baseline, almost twice as fast on shrunk down GPT like architecture. When trained for the same number of epochs, we see a substantial improvement in the pre-training performance, equivalent to a much larger architecture. The black vertical line denotes the epoch at which our architecture achieves the same performance as our baseline architecture.

$E/2$ transition from faster to slower movement. This approach enables the attention mechanism to utilize multi-scale features at varying rates across all layers and tokens, as explored in the next section. Further, these multi-scale structure can be made learnable, driven by just the next token prediction.

## 4 EXPERIMENTS

We explain how we incorporated the idea of infusing wavelets into a large language model pre-training. All of the models are trained from scratch, which required substantial computing. The main aim of these experiments is to show how the performance of the models across three modalities improves with/without doing intermediate modifications on embeddings. We also benchmark on LRA tasks.

### 4.1 BASELINE AND TRAINING SETUP

Our experiments based on the GPT-2 architecture, feature a stack of 10 Transformer decoder layers with a context length of 512, pretrained from scratch. Each modality—text, symbolic music, and raw waveform—shares the same architecture, using an embedding dimension of 128, a feed-forward dimension of 512, and 8 attention heads. We implement a two-layer feed-forward MLP within the Transformer, each layer matching the feed-forward dimension, rather than the single layer typical in Vaswani et al. (2017). The final decoder outputs to a dense layer of 2048 neurons, followed by a layer matching the vocabulary size: 27 for text8, 256 for raw waveform (Goel et al., 2022; Verma, 2022), and 388 for symbolic music. Baseline models consist of standard Transformer decoder blocks without modified embeddings. For our proposed architecture, we retain half of the embedding coordinates and impose either a fixed or learnable multi-scale structure on the other half for all intermediate layers. We do not compare against larger architectures, as this paper focuses on pre-training from scratch. Instead, we present a scaled-down version of GPT-2 suitable for resource-limited academia, evaluating pre-training performance with and without wavelet-inspired blocks. All models were trained from scratch in TensorFlow Abadi et al. (2016) for 25 epochs, starting with a learning rate of 3e-4, decreasing to 1e-5 when loss plateaued. Each model utilized 1M training points, totaling 500 million tokens, randomly cropped from the dataset. The MLP and attention layers used a default dropout rate of 0.1, with no additional regularization. We measured performance using negative log-likelihood loss, as this method improves the core architecture of the transformer-based GPT - helping achieve the objective we want to achieve: predict the next token correctly. Since we are operating on intermediate embeddings, our work can hopefully generalize to setups with structured data similar to text, raw audio, and symbolic music, where one can go from a fine-grained structure to a coarse structure. As shown in Figure 3, we can impose a multi-scale structure that allows the attention mechanism to not only learn dependencies across various embeddings but also inject some information that can capture coarse and fine-grained structure into these embedding coordinates.

## 4.2 PERFORMANCE ON MODALITIES

We compared the performance of our baseline architecture across three modalities—text, symbolic music, and audio waveform—with and without wavelet-based intermediate operations. Results showed significant performance improvements in all modalities with the same number of training steps. To illustrate, a 0.04 decrease in validation loss is comparable to going from a 16 to a 64-layer model on text-8 dataset (papers-with code, 2024). As shown in Figure 4, our modified GPT architecture achieves this loss nearly twice as quickly in terms of training steps compared to the original model showing GPT-like architecture can indeed take advantage of the structure that we imposed on half of the embedding dimensions. This speedup, i.e., the number of epochs/steps taken to achieve the same performance (SP: same performance epoch) is even smaller for raw audio, due to quasi-stationary nature of audio signals at smaller time scales (20-30 ms for harmonic sounds). For a sampling rate of 16KHz, a context length of 512 would correspond to 32ms, which may be one of the reasons that some of the coordinates nail down the contents of the context in fewer coordinates onto which we impose structure. The convergence is significantly faster for the raw waveform LLM setup, and achieving nearly twice the speed of text-8 and symbolic music. We also compare the absolute clock run times of our modifications in both learnable/non-learnable setups. In Table 1, we report the time taken to complete one epoch relative to our baseline architecture. Our method is computationally inexpensive, as it primarily involves fixed kernel multiplication or learning a single filter convolutional kernel with variable context lengths across different embedding dimensions.

Table 1: Comparison of the negative-log likelihood (NLL) scores for our architecture with three modalities with/without adding wavelet-based hierarchical structure and learnable wavelet transform.

| Modality | Baseline | Proposed | SP Epoch | SpeedUp | Relative GPU Hours |
|---|---|---|---|---|---|
| Text-8 | 0.93 | 0.92 | 14.5 epochs | 42% | 1.013 |
| Raw Audio | 1.84 | 1.70 | 3.7 epochs | 85% | 1.042 |
| Symbolic Music | 2.08 | 2.02 | 13 epochs | 48% | 1.059 |
| Text-8 (Learnable) | 0.93 | 0.91 | 12.9 epochs | 48.4% | 1.094 |
| Wiki-103 (Learnable) | 4.11 | 4.05 | 9.5 epochs | 62% | 1.130 |

## 4.3 SIMILARITIES AND DIFFERENCES WITH EMA

We compare with Exponential Moving Averages (EMA) on intermediate signals. Unlike Haar wavelet which takes fixed window weights, which takes the mean of the signal in the window, EMA uses exponential kernel. Let the signal $x_i^l(t)$, after the $l^{th}$ layer, be of length equal as context length, with $t$ being the token index from 0 to $L$ at embedding dimension $i$. The modified signal $s_t$ is:

$$s_0 = x_i^l(0) \quad s_t = \alpha x_i^l(t) + (1 - \alpha)s_{t-1}$$

where $\alpha$, the decay factor, satisfies $0 < \alpha < 1$. Unlike an EMA, our method uses a finite kernel, with zero weights outside a specified length, capturing multi-scale information. In text-8 experiments, we applied EMA on half of the embedding dimensions, with $\alpha$ linearly varying between 0 and 1 for dimensions 64 to 128. This under-performed compared to our baseline, with NLL score of 0.94, while our baseline and proposed method achieved scores of 0.93, 0.92, and 0.91 for non-learnable and learnable cases, respectively. Our method provides a simple, signal processing-based scheme, optimizing weights across multiple resolutions driven by next token prediction, outperforming EMA. Depending on $\alpha$, EMA filter produces an exponential kernel, while we maintain a constant kernel or allow weights learned from scratch optimized for the next token prediction. Further, EMA is an Infinite-Impulse Response (IIR) filter, whereas Haar wavelet based kernel is Finite Impulse Response (FIR) filter. Consequently, for each value update, the contributions from previous samples never reach zero. These can accumulate significantly at longer context lengths for certain $\alpha$. The recursive non-learnable nature of EMA IIR filter always ensures some contribution from all embeddings which may explain the performance degradation, whereas our method uses zero weights outside the kernel length, effectively capturing multi-scale information. We explain more in the Appendix.

## 4.4 EFFECT OF DEPTH AND MODEL DIMENSION

We explore two variants of our architecture for experiments on text-8 – i) reducing model dimension from 128 to 32 ii) reduce the number of layers. For the model with dimension 32 for a 10-layer

Transformer decoder architecture with eight heads, it still retains faster performance as a baseline, almost twice as fast as seen in Figure 4, and achieves the performance without doing the modification (as seen as baseline) around ten epochs. For the second experiment, we retain the exact architecture as reported in Table 1. We have 6 Transformer Decoder layers, keeping the rest of the parameters the same (feed-forward dimension four times that of the model dimension, eight attention heads) to see the effect of depth. The model, with Haar inspired modifications, similar to Table 1 results continues to get same performance as baseline twice as fast. Both of these experiments are shown in Figure 3.

### 4.5 MAKING MULTI-SCALE KERNELS LEARNABLE

We allow each of the kernels to be learnable. In the previous section, we defined the shape of the kernel, and computed approximate signals of intermediate layer activations across all layers, with different resolutions occurring at different embedding dimensions to mimic a causal version of wavelet transform. Now we allow each kernel of length $L$ at a particular level to be learnable for computing the *approximate signal* for various resolutions, a yet another way to compute it. By making the computation of approximate signal learnable, the model is able to learn how to weight every dimension of every decoder layer as opposed to putting a fixed kernel e.g. exponential weighted average. This as can be seen Algorithm 1 only allows 0.02M (20k) extra parameters to our base decoder architecture. This further improves our performance from 42% to 48% faster speedup to get a similar baseline performance, seen in Figure 4, carried out on the text-8. We also benchmark on Wiki-103 to demonstrate that our method works with the GPT-2 tokenizer. As shown in Table 1, we match the performance of a 10-layer architecture at more than twice the speed. In addition to faster convergence, we see a 3.6-point improvement in perplexity scores over the baseline model. While our architecture, with a 512 context length and 128 model dimension, is a simplified version of GPT-2/3, constrained by academic resources, Section 4.4 shows it scales with model size, highlighting its potential for future improvements for decoder only LLM architectures across modalities and datasets.

## 5 LONG RANGE ARENA BENCHMARKS

We adapt our architecture for Long-Range Arena (LRA) tasks Tay et al. (2021), which test models on long-range prediction across text, images, and mathematical expressions. These tasks evaluate the model's ability to handle similarity, structure, and reasoning over extended contexts. We focus on transformer-based architectures, as recently reported by Liu et al. (2024), while other variants include state-space and hybrid models or tweaking attention mechanism. For text, we perform binary classification on the IMDb review dataset (Maas et al., 2011) using byte-level data with a context length of 2048 to determine if a movie review is positive or negative. For images, we use CIFAR-10 from the LRA benchmark, classifying sequences of 3072 pixels into one of ten categories. Lastly, we benchmark on Long ListOps, testing the model's ability to understand hierarchically structured data in extended contexts. As per LRA paper Tay et al. (2021), "The dataset is comprised of sequences with a hierarchical structure and operators MAX, MEAN, MEDIAN and SUM_MOD that are enclosed by delimiters (brackets). An example (much shorter) sequence is as follows: **INPUT:** [MAX 4 3 [MIN 2 3] 1 0 [MEDIAN 1 5 8 9, 2]] **OUTPUT:** 5. In our task, we use a version of ListOps of sequence lengths of up to 2K to test the ability to reason hierarchically while handling long contexts. In the above example, the model needs to access all tokens and model the logical structure of the inputs to make a prediction. The task is a ten-way classification task and is considerably challenging." We use the setup provided by Khalitov et al. (2022) to extract the data and be uniform with other benchmarks. We use a nearly identical architecture for all three modalities, only modifying the embedding matrix to accommodate different tokenizers and output categories. Our baseline consists of a 6-layer causal Transformer decoder with a model dimension of 32 and a feed-forward dimension four times that of the embedding dimension. We extract the last token of the sequence as a 32-dimensional embedding for classification, followed by a dense layer with 2048 neurons and another dense layer corresponding to the number of categories. The input goes through an embedding layer that converts discrete tokens into a 32-dimensional vector. The input vocabularies are 256 for text and image, and 16 for ListOps. The context lengths are 2048, 3072, and 1999 tokens, respectively, with output categories of 2, 10, and 10. In our modified architecture, we introduce our waveletGPT module between each decoder layer, retaining half of the embedding dimensions as they are. For the other half, we use non-learnable kernels, increasing the kernel size from 2, 4, and 8 to 512 linearly for dimensions 16 to 32, while maintaining the causality assumption. This introduces

Table 2: Performance on LRA tasks (Tay et al. (2020b)) as reported in Liu et al. (2024). Bold indicates the best-performing model, underlined indicates the second best. We use a baseline architecture for all three datasets (Section 5) and modify intermediate embeddings by imposing a hierarchical structure. Non-transformer based, modified attention based or hybrid architectures are not reported.

| Transformer Based Attention Models | ListOps | Text | Image |
|---|---|---|---|
| Transformer (Vaswani et al., 2017) | 36.37 | 64.27 | 42.44 |
| Local Attention (Tay et al., 2020b) | 15.82 | 63.98 | 41.46 |
| Linear Trans. (Katharopoulos et al., 2020) | 16.13 | 65.90 | 42.34 |
| Linformer (Wang et al., 2020) | 35.70 | 53.94 | 38.56 |
| Sparse Transformer (Child et al., 2019) | 17.07 | 63.58 | 44.24 |
| Performer (Choromanski et al., 2021) | 18.01 | 65.40 | 42.77 |
| Sinkhorn Transformer (Tay et al., 2020a) | 33.67 | 61.20 | 41.23 |
| Longformer (Beltagy et al., 2020) | 35.63 | 64.02 | 40.83 |
| BigBird (Zaheer et al., 2020) | 36.05 | 64.02 | 40.83 |
| Luna-256 (Ma et al., 2021) | 37.25 | 65.78 | 47.86 |
| Reformer (Kitaev et al., 2020) | 37.27 | 56.10 | 38.07 |
| FNET (Lee-Thorp et al., 2022) Non-Causal | 37.27 | 56.10 | 38.07 |
| WavSPA – Ada Transformer (Zhuang et al., 2024) - Non-Causal | 55.40 | **81.60** | 55.58 |
| Ours (GPT Baseline With Classification Head) | 41.65 | 65.32 | 49.81 |
| **Ours (WaveletGPT With Classification Head)** | **57.5** | 66.38 | **59.81** |

highways that hierarchically process data at each embedding and Transformer decoder layer without adding parameters, similar to our approach for LLM. As shown in Table 2, we achieve notable gains across all three modalities, where even small improvements are worth reporting. We significantly outperform non-causal methods, such as (Zhuang et al., 2024), with nearly 2% improvement on ListOps and 4.5% on a much smaller architecture—ours has 32 dimensions and six layers compared to 128 dimensions and eight layers. We limit our comparison method for fairness only with vanilla Transformer architectures. We also compare with two non-casual architectures that incorporated signal processing based ideas: FNET and WavSPA. We do not compare it with other sophisticated state space based methods or complex architectural changes as it would have required further tuning to our method/ sigbnificant architectural changes than straightforward simple tweaks to have a fair comparison. Compared to non-causal FNet, our model significantly outperformed all three LRA tasks, achieving 20% improvement on ListOps and Image and 10% on text. The most notable gain is in the ListOps task, which involves modeling a hierarchical, tree-like structure of math operations, making our model particularly suitable. To the best of our knowledge and Liu et al. (2024), this is the best performance achieved by a simple attention-based Transformer architecture on LRA tasks.

## 6 CONCLUSION AND FUTURE WORK

We showcase the powerful incorporation of a core signal processing idea, namely wavelets, into large language model pre-training. By imposing a multi-scale structure onto every intermediate embedding, we achieve the same performance 40-60% faster, compared to a baseline architecture. We achieve a substantial performance boost if we train for the same number of steps. Our method generalizes across three modalities: raw text, symbolic music, and raw audio, giving similar performance speedups. Several exciting directions can be explored in future work, including incorporating more advanced ideas from wavelets and multi-resolution signal processing onto large language models. It will be interesting to see how the model behaves for different variants of multi-scale structures.

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

## A    REPRODUCIBILITY STATEMENT

We have included details of the dataset information and pre-processing pipelines, all publicly available to reproduce our results. Further, we have explained our algorithm, including all the necessary architectural information, learning rate schedules, algorithm details, training times, etc, to reproduce the results. Further, we will open-source our model code upon acceptance.

## B    ETHICS STATEMENT

No human subjects were used in this study. We aim to reduce the amount of time taken to pre-train an LLM. This paper is concerned with improving LLM pretraining and boosting its performance. So, all ethical concerns corresponding to large language models are identical. We do not open-source our code at this time but will do so upon acceptance of the paper.

## C    COMPARISON WITH EXPONENTIAL MOVING AVERAGES

We compare our method with Exponential Moving Averages (EMA) on the intermediate signals. This is widely used in time-series analysis for smoothening data, and it is another type of way that can modify intermediate signals. We proposed Haar wavelet, a multi-resolution kernel that can look at the input signal at various levels of scales depending on embedding dimension. We will now compare it against an EMA baseline and motivate where we differ and are similar to our proposed method.

## C.1 BACKGROUND

Loosely speaking, instead of a moving average filter taking the mean of the signal, an EMA uses a different kernel, i.e., an exponential function. Meanwhile, a moving average kernel assigns equal weight to all time points. If we assume that the $x_i^l(t)$ signal of length is equal to context length after the $l^{th}$ layer with $t$ being the token index going from 0 to context length $L$ at embedding dimension $i$, we can define the modified exponential smoothed version of the signal $s_t$ as

$$s_0 = x_i^l(0) \quad s_t = \alpha x_i^l(t) + (1 - \alpha)s_{t-1}$$

Where $\alpha$ is the decay factor, it always satisfies $0 < \alpha < 1$. We can observe that for each of the tokens, depending on the decay factor $\alpha$, we assign weights to the more recent values over the past values. When $\alpha = 1$, the weightage given is only to the current observation, and when $\alpha = 0$, it is just flat and gives equal weightage. The differences with moving average filters are evident i.e., first, the moving average filter gives equal weight to all of the values in a window to update the values of a particular window. Depending on the value of $\alpha$, an EMA filter gives an exponential weighted kernel. However, from the definition itself, an EMA filter, irrespective of the value of $\alpha$, is an Infinite-Impulse response (IIR) filter, whereas a moving average filter is a finite impulse response (FIR) filter. Therefore, for every value update at a particular location, the values of dependencies of the previous samples will never be zero and relatively small. One can see that these values can add up significantly for some values of $\alpha$ when we are predicting the next tokens at longer context lengths. Due to the nature of the IIR filter, the values are never zero. They are assigned values weighed depending on the previous observation as $1, 1 - \alpha, (1 - \alpha)^2, (1 - \alpha)^3,...$

On the other hand, our proposed method includes wavelets composed mainly of FIR filters, including Haar or Daubechies. They are, therefore, only limited to a finite duration and can be adapted in multi-resolution setups with varied window lengths, as we have proposed in our paper. This allows us to have multi-scale information where we look at any signal at different resolutions with varied window lengths, with no contributions from components outside the desired window. (as we set the contribution from those values as 0). EMA, on the other hand, would still have some contribution from every component due to its recursive nature. One could also have a version similar to our method where one could vary $\alpha$ depending on the embedding dimension $i$. The update equations would now be a function of $i$, i.e.

$$s_0 = x_i^l(0) \quad s_t = \alpha_{(i)}x_i^l(t) + (1 - \alpha_{(i)})s_{t-1}$$

This would introduce different dimensions decaying at different rates. Even with varying decay rates, because of the inherent nature of the IIR filter, we still give weightage to all values, which are never zero, unlike the FIR filter, which utilizes a window and gives no weightage to values outside the window.

Training all possible values of $\alpha$ is beyond our scope and resources. We, therefore, give the best equivalent of the EMA algorithm with our proposed method, as described in the next section.

## C.2 EXPERIMENTS AND RESULTS

We retain our baseline architecture precisely the same for text-8. We train for a context length of 512 with the same setup reported in our baseline section and the same dataset, with the only tweak being taking the baseline architecture and adding an EMA layer to it. We choose the number of decoder blocks to be 10, with 128 as the embedding dimension, the feed-forward dimension to be 512, and the number of heads to be 8. We opt for a two-layer feed-forward MLP inside the Transformer block after the attention block instead of a single layer typically used in Vaswani et al. (2017), with both the layers sharing the same number of neurons, i.e., 512, that of the feed-forward dimension. The final output layer of the Transformer decoder is then followed by a dense layer of 2048 neurons, followed by a dense layer of the same size as the vocabulary. This vocabulary size varies in the three modalities. For text8, it is 27, which is the number of characters plus an added extra token for space. Similar to our proposed method, we experiment with keeping half of the embedding dimensions in all the layers the same without any modifications. For the other half of the embedding dimension after all layers, we carry out EMA on 1-D signals, as described in the previous section, with $\alpha$ varying

from 0 to 1 linearly for embedding dimensions 64 to 128. We see a drop in performance compared to our baseline architecture and achieve an NLL score of 0.94. For comparison, our baseline trained on text-8 scored 0.93, with our proposed method being 0.915 and 0.91 for learnable and non-learnable cases, respectively.

## C.3 DISCUSSION

There can be many reasons why EMA degrades performance. One of them can be tuning $\alpha$. There can be many possible choices, and tuning them for an expensive LLM pretraining is tough. Our proposed method, WaveletGPT, on the other hand, has a simple way of giving the weightage, which is grounded in signal processing and outperforms EMA smoothening. Further, in our learnable section, the architecture can learn the optimal **weights** in which, depending on the space spanned by the intermediate signals found inside LLM, it learns weights from scratch at different resolutions from the finest, i.e., window length 1 to the coarsest, i.e., window length as the context length 512.

