# OpenReview forum: "WaveletGPT: Wavelet Inspired LLMs"
_ICLR.cc/2025/Conference — Submitted to ICLR 2025_

### Official Review · Reviewer_Tyuw · 2024-10-28

**Soundness:** 2
**Presentation:** 1
**Contribution:** 2
**Rating:** 1
**Confidence:** 3

**Summary:**

This paper proposes WaveletGPT, which encompasses incorporating wavelet transform within the intermediate representations to impose multi-scale learning. Instead of going for SOTA, the paper focuses on demonstrating how the new WaveletGPT stands compared to standard Transformer based approaches, albeit on reduced scales.

**Strengths:**

The paper is sufficiently original in ideation, but the quality and clarity are not. The subject matter itself had the potential to have a significant impact if investigated well.

Some pros of the paper are:

- Several domains are covered: text, raw audio and music, which is great, since the approach has the potential for impacting multiple domains.
- The approach sounds interesting and demonstrates faster convergence to same NLL score compared to standard transformer based baseline.

**Weaknesses:**

The paper has the following shortcomings.

1. The paper is tremendously hard to read. The introduction reads like statements stitched together and is incoherent. Sections of the introduction are repeated, for instance: lines 76-79 about clockwork RNN. Related works are explained rather poorly. This paper needs to be proofread and phrased better: writing alone is unfit for an ICLR publication.
2. The datasets used are rather arbitrary and their usage is motivated well enough. Why not use a standard verifiable audio task such as speech?
3. Dataset scale. Even if we let the choice of "text-8" slide, Youtube-Mix-8 and Maestro are not exactly large datasets, with fewer than 10k samples each.
4. The inclusion of LRA tasks is great, but there is simply not enough evaluation done in the paper. WaveletGPT should be investigated on more mainstream, even if smaller scale, datasets and benchmarks. The choice of architecture is fine, but more evaluations are needed.
5. Table 2 states that "non-transformer based or modified attention based hybrid architectures are not reported", however, FNET and WavSPA makes the cut. The choice of what models were benchmarked/reported is not justified enough.

The idea is interesting, but simply not executed well enough. More empirical evaluations are warranted, and a significant overhaul in the writing department is needed as well.


POST-REBUTTAL EDIT

Updated some language that the authors had issues with.

**POST-REBUTTAL EDIT 2: responses to the final comment made by the authors**

>  The goal in this work is only pretraining performance and not going after NLP benchmarks that are often used to evaluate much much larger models.

> We have in fact gone to show the same architecture works for raw pixels, raw audio samples, math expressions, byte level text, GPT-2 tokenized text, character level text, acoustic tokens and audio waveform which is a more comprehensive result in our opinion as compared to a larger NLP dataset

Yes, but you have gone into that now, after the rebuttal. Pre-rebuttal the authors used 2 small, outdated NLP datasets and 2 small audio datasets.

---

> We again differ and it is not a correct statement. We are modeling raw audio therefore the tokens here are raw samples. Your statement is incorrect.

That still doesn't change the fact that 5 hours of audio is a small audio dataset. Numerous approaches work with raw audio waveforms. For eg. wav2vec2 processes 16 kHz raw waveforms from LibriSpeech and LibriVox, which are ~1k hours and 60k hours of speech audio data, respectively. I'm not asking the authors to experiment on 60k hours of speech data, but my statement that the datasets used in the study are small is not incorrect.

> The dataset is large from point of view of raw acoustic sample tokens.

From the point of view of total hours of audio data, these datasets are small. Yet the authors are adamant in claiming that my statements are incorrect. What makes this discourse an even bigger exercise in futility is that the authors did conduct additional experiments on larger audio datasets: viz, LibriSpeech and FSD50K. The authors should be amplifying the improvements they've made and commenting and clarifying the additional analysis they've done (there is next to no information about how you did these experiments, apart from the fact that you show us some numbers), not trying to assert that the reviewer is wrong.

>  The same argument can be made for MAESTRO which was used to train Music Transformer models the most cited music generation paper.

This paper is not a music generation paper.

---

Thanks for posting the FSD50K and LibriSpeech results, but there's no accompanying text for these experiments? You're mixing metrics in the table. There's no information about how these models were trained.

Also, I would like to note that the errors and issues with writing still persist and have not been updated in the main paper.

---

> To clarify, these were not allegations but direct quotes from the review including the new statements and opinions like 'It is disrespectful and undermining of the time and value reviewers put into reviewing your paper out of their schedule' and 'an appeal to established authority.' We felt the language and inappropriate statements are not suitable in the context of public reviews.

I'm not going to comment on this any further, apart from saying that I am completely in the right to point out what felt was undermining and disrespectful of the time and effort I have put into reviewing this paper and writing prompt, detailed responses to authors comments in a public discourse.

---

> My main concern is that the authors use the argument ...... we will not conduct so many additional experiments in such a short rebuttal period." I think such practices are contrary to the spirit of ICLR.

I agree with reviewer 4yst here.

---

Response to Last rebuttal comments

> We mentioned the link where the paper is updated. We share it again. The updated paper is here with extra experiments. https://drive.google.com/file/d/1u9OGe6VwnKhkyIBs9-k5a87c-rVFF637/view?usp=sharing

I can only make review decisions for paper revisions as they appear on the ICLR submission page. Maybe the Chairs can suggest otherwise, but sharing an external link to your updated paper allows post-rebuttal editions to the paper and also prohibits tracking of changes made to the paper, which to the best of my knowledge would be against ICLR policy.

Regarding other comments:

- The authors bring up SaShiMi, an audio generation paper. Evaluating the long-term context modelling capabilities of your approach through YouTube-Mix-8 is fine, but the dataset alone was not sufficient to evaluate your paper because your paper is not a music generation paper, it's an LLM paper, as the authors have repeatedly said so.

- The authors are getting fewer tokens for LibriSpeech than YouTube-Mix-8 because of design decisions and token sampling strategy. LLMs operate on tokens, fair, but that doesn't make the number of hours in an audio dataset irrelevant.

---.

**Questions:**

N/A

---

> ### Author Response · Authors · 2024-11-28
> **Response 1**
>
> Dear Reviewer Tyuw
>
> You have been made a reviewer in the top-most ML conference perceived by the researchers, and you are making these comments publicly. What kind of language is this ? You are completely in your right to give any score you deem fit. But what it does not give you is the right to be unprofessional, dismissive and detracting.
>
> What is not right is to use words like "what a shame" twice, which is not a professional language in top conferences like ICLR or tone and use of dismissive language like "unfortunately that's where the pros end".
>
> This is a serious violation of the reviewer guidelines in any conference and I hope program chairs, senior area chairs and area chair make a note of this.
>
> The updated paper is here with extra experiments. https://drive.google.com/file/d/1u9OGe6VwnKhkyIBs9-k5a87c-rVFF637/view?usp=sharing
>
>
> Weakness 1: We agree that the paper is hard to read. We have removed the sections that are repeated related to clockwork RNN. The related work is inclusive of all possible approaches related to our work that were possible from different angles such as distillation. We have ran the paper through spell checks and automated writing tools.
>
> Weakness 2: We have added LibriSpeech corpus to do the same now and reported speedup and the results. We strongly disagree on this about the dataset. Wiki-103 and text-8 are one of the most widely cited datasets for language modelling across thousands of paper. Further Long-Range Arena benchmark is also one of the most popular dataset and metric used to explore the effect of long-context modelling besides language modelling. We do want to make it clear here that the goal here was to showcase that the method generalizes across text i.e. characters, byte-pair tokens as in GPT-2, speech tokens as derived from ENCODEC as well as raw audio waveform. Further we also have now added the results for Audio Classification benchmark FSD-50K which shows that the method is generic enough for non-token based input representation e.g. raw audio waveform.
>
> Weakness 3: This is an incorrect statement. On the contrary Youtube-Mix-8 and Maestro are one of the largest datasets available out there. Youtube-Mix-8 contains 5 hours of piano music sampled at 16kHz would give us 3600 sec x 16000 samples x 5 hours ~ 300 million tokens. MAESTRO contains about 200 hours of piano music and contains on the order of 50-100 million tokens. Despite of all of this we have oversampled and clearly stated in our paper that we use about 500 million tokens to account for prediction of the same tokens with different context (in the beginning or the end)
>
> Weakness 4: These are mainstream datasets and benchmarks. To give an example, a paper called MegaByte just reported NLL score only in the entire paper was written by respected authors in a well renowned ML venue similar to ICLR. We have taken the datasets that have been used by a lot of papers, and benchmarks reported by several academic institutions. Further, we have clearly stated that this paper was written in an academic setup with limited resources and that should be taken into consideration for limited yet convincing experimentation.
>
> Weakness 5: We have made it clear now in our writeup that FNET and WavSPA are signal processing based method and hence we have included them.

---

> ### Comment · Reviewer_Tyuw · 2024-11-29
> **Response to the authors, Part 1/2**
>
> Dear authors,
>
> The language of the review has been updated. Apologies, but the intention was not derogatory in any way. I will first address the facts posted in the author rebuttal:
>
> > Weakness 1: We agree that the paper is hard to read. We have removed the sections that are repeated related to clockwork RNN.... We have ran the paper through spell checks and automated writing tools.
>
> Thanks for acknowledging that. However, the associated pdf of the paper still has the same mistakes and I am not in a position to conclude whether these changes were indeed made or not.
>
> ----
>
> > Weakness 2: We have added LibriSpeech corpus to do the same now and reported speedup and the results.
>
> I do not see any additional results on LibriSpeech in the main body of the paper, nor the appendices, nor in any comment posted by the authors. Can the authors ascertain that they have indeed added these results?
>
> > Wiki-103 and text-8 are one of the most widely cited datasets for language modelling across thousands of paper. Further Long-Range Arena benchmark is also one of the most popular dataset and metric used to explore the effect of long-context modelling besides language modelling.
>
> Weakness 2 was about motivating the usage of the datasets that you did use: there is a rather small section (Sec 2) of about 20 lines that is dedicated to the same, but this falls short.
>
> text-8 is well-cited dataset sure, but it cannot be the sole basis of demonstrating that your approach will scale suitably for training larger scale LLMs. It has only 100 million characters, it doesn't even have punctuations or upper case letters or numbers. It was obtained from a pretty old version of english wikipedia and lacks diversity and domain coverage that modern LLMs need to keep up with.
> Simply put, text-8 is great for benchmarking and quick prototyping, but it is not sufficient to evaluate whether your model captures high level semantics and is suited for taks that need richer language understanding. wiki-103 shares a lot of these drawbacks as well, and it is not clear from the paper why wiki-103 and text-8 together constitute a sufficient NLP modelling benchmark.
>
> > Further we also have now added the results for Audio Classification benchmark FSD-50K which shows that the method is generic enough for non-token based input representation e.g. raw audio waveform.
>
> Again, I do not see any of these additional results anywhere.
>
> ---
>
> > Weakness 3: This is an incorrect statement. On the contrary Youtube-Mix-8 and Maestro are one of the largest datasets available out there. ... with different context (in the beginning or the end)
>
> The comment was made from the perspective of the number of samples, scale and coverage. MAESTRO and Youtube-Mix-8 **are** small audio datasets. Sampling 300 million tokens from a dataset with 5 hours of total audio data does not make the dataset large, so the authors missed my point. By that argument, a dataset composed of hi-resolution DSD music files with an hour worth of music will yield even more tokens: that does not mean it's a "larger" dataset than MAESTRO and Youtube-Mix-8 or that it offers better domain coverage!
>
> In the rebuttal, the authors have stated that they have done experiments on LibriSpeech and FSD50K, which I acknowledge would be a great step towards addressing this concern, because it goes beyond modelling long range context alone, and ventures into modelling semantic information on a large scale. I can't find any of these experiments yet, but I'm more than happy to revisit my score if the authors can point me to them.
>
> ---
>
>
> (continued..)

---

> ### Comment · Reviewer_Tyuw · 2024-11-29
> **Response to the authors, Part 2/2**
>
> > Weakness 4: These are mainstream datasets and benchmarks.... Further, we have clearly stated that this paper was written in an academic setup with limited resources and that should be taken into consideration for limited yet convincing experimentation.
>
> My original review commended the addition of LRA benchmarks, and we can assure the authors that the reviewer is well aware of the benefits of LRA.
> Also, the original review did not raise questions about NLL score reporting, so I do not know why the authors felt the need to say that reporting NLL score alone in some other paper was sufficient. I never raised that question.
>
> Given what the author wrote in their response, I would like to point out that the paper currently under review is this paper, here, in front of us. What another paper written by reputed authors in another reputed ML conference did or did not is none of my concern: every paper is evaluated within it's own context. What is reasonable within the context of another paper is not always appropriate for another. What the author wrote sounds like an appeal to established authority: that X reputed authors did Y and got accepted to a similar level of conference, so it must be right. That is not grounds for evaluating what this paper has presented. The paper pointed out by the authors, Megabyte, evaluates larger scale models using several large-scale datasets that appropriately evaluate the approach. This paper, in it's original form, does not.
>
> > Further, we have clearly stated that this paper was written in an academic setup with limited resources and that should be taken into consideration for limited yet convincing experimentation.
>
> And the reviewer listened and understood, hence not questioning the usage of the GPT-2 architecture, nor questioning the absence of SOTA performance, but that does not change the fact that the evaluation was not sufficient.
>
> Again, the authors have stated that they have done more evaluations on LibriSpeech and FSD50K, which would be a great step forward in addressing this concern. But I can't find any of these anywhere.
>
> ---
>
> > Weakness 5: We have made it clear now in our writeup that FNET and WavSPA are signal processing based method and hence we have included them.
>
> Ok, thanks. The section where you cover these approaches is difficult to understand. For example, lines 92-94 describing WavSPA is unclear. If updated, I'll consider this issue resolved.
>
> ---
>
> **Overall**, the authors claim that they have made some writing changes, adding additional experiments and evaluations. At the time of posting this comment, I can't see these changes anywhere, but I'll be more than happy to reconsider my score once I do.
>
> ---
>
> Now that I have addressed the responses that the authors made in response to the scientific elements of my review, I would like to address the other comment:
>
> Dear authors. I apologize for using the word shame in the review, the objective was not to undermine your work. However, I must point out that you have also submitted a paper to a top-ML conference. Reviewers are spending valuable time out of their schedule to review your paper for no personal gain, solely for the sake of science. Submitting what seems to be a rushed, difficult-to-read paper that was not proofread well enough, followed by making allegations that the reviewer "dismissed" your paper after they spent hours trying to comprehend your work and do it justice is also unprofessional. It is disrespectful and undermining of the time and value reviewers put into reviewing your paper out of their schedule.

---

> ### Author Response · Authors · 2024-12-03
> **Response to Tyuw 2nd cycle**
>
> | Modality                | Baseline | Proposed | SP Epoch     | SpeedUp | Rel. GPU Hrs |
> |-------------------------|----------|----------|--------------|---------|--------------|
> | Text-8                 | 0.93     | 0.92     | 14.5 epochs  | 42%     | 1.013        |
> | Raw Audio              | 1.84     | 1.70     | 3.7 epochs   | 85%     | 1.042        |
> | Symbolic Music         | 2.08     | 2.02     | 13 epochs    | 48%     | 1.059        |
> | Text-8 (Learnable)     | 0.93     | 0.91     | 12.9 epochs  | 48.4%   | 1.094        |
> | Wiki-103 (Learnable)   | 4.11     | 4.05     | 9.5 epochs   | 62%     | 1.130        |
> | LibriSpeech (Learnable)| 2.43     | 2.40     | 9.2 epochs   | 63.2%   | 1.110        |
> | FSD-50K                | 40.6%    | 42.8%    | 32/92 epochs | 65.2%   | 1.037        |
>
>
> ** text-8 is well-cited dataset sure, but it cannot be the sole basis of demonstrating that your approach will scale suitably for training larger scale LLMs.
>
> Everything is relative in terms of compute available when we say text-8 and wiki-103 are great for benchmarking and prototyping. It is not like modern LLMs architecture fail to keep up with diversity and domain.... it is all relative to the compute and parameters. We have in fact gone to show the same architecture works for raw pixels, raw audio samples, math expressions, byte level text, GPT-2 tokenized text, character level text, acoustic tokens and audio waveform which is a more comprehensive result in our opinion as compared to a larger NLP dataset. The goal in this work is only pretraining performance and not going after NLP benchmarks that are often used to evaluate much much larger models.
>
> ** MAESTRO and Youtube-Mix-8 are small audio datasets
>
> We again differ and it is not a correct statement. We are modeling raw audio therefore the tokens here are raw samples. Your statement is incorrect. There is around 5 hours of audio which is about 5 hours x 16000 samples/sec x 3600 seconds /hr ~ 288 million tokens present in the dataset. The dataset is large from point of view of raw acoustic sample tokens. The same argument can be made for MAESTRO which was used to train Music Transformer models the most cited music generation paper.  We have added the experiments for 1000 hours of Librispeech .
>
>
> About the apology: We appreciate your acknowledgment of the inappropriate language and tone in your reviews. To clarify, these were not allegations but direct quotes from the review including the new statements and opinions like 'It is disrespectful and undermining of the time and value reviewers put into reviewing your paper out of their schedule' and 'an appeal to established authority.' We felt the language and inappropriate statements are not suitable in the context of public reviews. Moving forward, we hope to maintain a constructive and professional discourse.

---

> ### Author Response · Authors · 2024-12-03
> **Last rebuttal comment to Post Rebuttal Edit 2**
>
> Dear Reviewer Tyuw,
>
> We mentioned the link where the paper is updated. We share it again. The updated paper is here with extra experiments. https://drive.google.com/file/d/1u9OGe6VwnKhkyIBs9-k5a87c-rVFF637/view?usp=sharing
>
> @AreaChairs, SeniorAreaChairs, Reviewers The comments have gone too far now, and I will respectfully and humbly correct any lies/facts that are not right that are present/that you mention.
>
> Reviewer Comment: "Pre-rebuttal, the authors used two small, outdated NLP datasets and two small audio datasets."
> Our Response: This is not true. In addition to "2 small outdated NLP datasets", text-8 and Wiki-103, each containing more than 100 million tokens, used MAESTRO, a large well-cited dataset on MIDI tokens, and YouTube-Mix-8, a popular benchmark as reported by "SasHIMI: Its Raw paper by Chris Re, et al." containing around 300 million tokens. In addition, in the pre-rebuttal phase, we have included the LRA benchmark, another popular dataset that had images at the pixel level, math expressions and byte-represented text. The goal was to include a text benchmark at the byte, character, and GPT-2 token levels. We have reported the results on four different modalities with the new addition of acoustic tokens, and in addition to that, we have reported on audio classification benchmarks. Further, we have reported on LRA benchmarks on three datasets and done a study on the effect of context length and the number of layers that were the most important. The paper has been tested on raw pixels, raw audio, byte text representation, GPT-2 tokens, characters, acoustic tokens, and math expressions, which shows the strength of our work.
>
> Reviewer Comment: "That still doesn't change the fact that 5 hours of audio is a small audio dataset. Numerous approaches work with raw audio waveforms. For, e.g. wav2vec2 processes 16 kHz raw waveforms from LibriSpeech and LibriVox, which are ~1k hours and 60k hours of speech audio data, respectively. I'm not asking the authors to experiment on 60k hours of speech data, but my statement that the datasets used in the study are small is not incorrect."
> Our Response: This, again, is an incorrect statement and inconsistent with the problem and the goals at hand. The hours are irrelevant. We are benchmarking LLM pretraining, so how many tokens are present in a particular representation matters. You pointed out that LibriSpeech satisfies the constraints of a larger dataset, and 5 hours of audio is a small dataset.
> In fact, the number of tokens we trained on 1000 hours of LibriSpeech on fewer tokens than that of YouTube-Mix-8. For LibriSpeech using ENCODEC tokens, 1000 hours corresponds to 1000 hours x 3600 seconds/ hour x 75 tokens/ sec = 270 million acoustic tokens, which is less than the number we used for YouTube-Mix-8, which is around 300 million.
> The LLM cares about tokens and representation. Hope this makes it clear now. The choice of Youtube-Mix-8 is also used in SaSHIMI by Chris Re and was the reason we chose to report the result, as it is used in several papers.
>
> Reviewer Comment: "I'm not going to comment on this any further, apart from saying that I am completely in the right to point out what I felt was undermining and disrespectful of the time and effort I have put into reviewing this paper and writing prompt, detailed responses to authors comments in a public discourse."
> Our Response: I again would like the senior area chairs, area chairs and program chairs to look into this. One just cannot continue to say, "The authors were disrespectful to the reviewers" by submitting a poor paper. The very first inappropriate comments about improper words that the reviewer apologized for using the term "what a shame " have continued throughout this rebuttal phase, with the reviewer being consistent with the incorrect tone, basic courtesy, and professionalism. Hope it can be looked into.

---

### Official Review · Reviewer_G6qn · 2024-11-01

**Soundness:** 3
**Presentation:** 1
**Contribution:** 4
**Rating:** 6
**Confidence:** 2

**Summary:**

This paper introduces a novel approach to Large Language Model (LLM) pre-training by embedding wavelet-based multi-scale structures into a GPT-style architecture. This method enables the model to access intermediate embeddings at different temporal resolutions without adding extra parameters, resulting in significant speedups (40-60%) in training time across text, raw audio, and symbolic music modalities. By structuring embeddings in a multi-scale manner, the authors achieve performance comparable to larger models, highlighting a new direction for improving model performance through internal structural modifications rather than sheer scale. The results suggest promising potential for incorporating signal processing techniques, such as wavelets, into LLM pre-training.

**Strengths:**

The proposed method is highly original, combining traditional signal processing concepts with contemporary LLM pre-training. The research methodology is robust, and the work addresses an important problem in LLM development, particularly in optimizing efficiency for multi-modal data without expanding model parameters.

**Weaknesses:**

* The paper lacks explicit details about model parameters; assuming a GPT-2 scale (~120M parameters), an exploration of 300M-1B parameters could be feasible given the authors’ computational resources. Such scaling would be valuable to determine if results hold at larger model sizes.
* Although the approach aims to address long-sequence modeling, experiments are limited to sequences of length 512.
* The paper lacks detailed downstream task results and analysis, limiting the assessment of its practical impact.  You may test on tasks like text summarization, question answering, or machine translation etc. to demonstrate the model's effectiveness in real-world applications

**Questions:**

The following questions are easier to solve under limited computational resources. I would change the decision to marginal accept if the answer to the following questions are intuitive.
* What implications arise from the significant loss reduction observed after applying wavelet transforms? For instance, could case studies illustrate improved sentence generation quality during inference? You can compare generated text samples from the baseline and wavelet-enhanced models on specific prompts or analyzing perplexity improvements on different types of text (e.g., formal vs. informal language).
* While the authors achieve better performance with learnable wavelet transforms, analyzing the learned wavelet parameters compared to traditional Haar wavelets would add valuable insights. It would be most interesting to visualize the learned wavelet shapes, compare their frequency responses to Haar wavelets, or analyze how they evolve during training.
* In Section 4, various experimental data are mentioned without tabular representation or inclusion in the appendix. Could the authors clarify the reasoning behind this omission?
* The paper includes two figures detailing Haar wavelet definitions yet lacks visual representation for the more critical learnable wavelet transform defined in Section 3.3.
* Many paragraphs, particularly in the introduction, are quite lengthy, with the introduction itself comprising only 1-2 dense paragraphs, making the paper somewhat difficult to read. Would breaking these up improve readability?

---

> ### Author Response · Authors · 2024-11-28
> **Response 1**
>
> Here are more experiments added for 1000 hours of LibriSpeech and Audio Classification on FSD-50K
>
> | Modality                | Baseline | Proposed | SP Epoch     | SpeedUp | Rel. GPU Hrs |
> |-------------------------|----------|----------|--------------|---------|--------------|
> | Text-8                 | 0.93     | 0.92     | 14.5 epochs  | 42%     | 1.013        |
> | Raw Audio              | 1.84     | 1.70     | 3.7 epochs   | 85%     | 1.042        |
> | Symbolic Music         | 2.08     | 2.02     | 13 epochs    | 48%     | 1.059        |
> | Text-8 (Learnable)     | 0.93     | 0.91     | 12.9 epochs  | 48.4%   | 1.094        |
> | Wiki-103 (Learnable)   | 4.11     | 4.05     | 9.5 epochs   | 62%     | 1.130        |
> | LibriSpeech (Learnable)| 2.43     | 2.40     | 9.2 epochs   | 63.2%   | 1.110        |
> | FSD-50K                | 40.6%    | 42.8%    | 32/92 epochs | 65.2%   | 1.037        |
>
>
>
> Thank you for liking our work. Yes, we agree that combining traditional signal processing concepts would open a whole lot of tools for LLMs to be improved in efficiency. We have added more experiments in the paper now on acoustic tokens of LibriSpeech as well as a well reported audio classification benchmark.
>
> Weakness 1: This was written in academic setup with limited compute. That being said we have tried to address the effects of lower number of layers and the smaller model dimension and it seems that results continue to hold. We can only speculate what happens in the regime of 300M-1B and it could go either way, if we were on to play devils advocate. But we are confident that our method would scale.
>
> Weakness 2: That is not true. For Long Range Arena benchmark the context length for three datasets are 1999, 2048 and 3172 which shows that our method can adapt to long sequence modelling.
>
>
> W3: We beg to differ on this. These are mainstream datasets and benchmarks. To give an example, a paper called MegaByte just reported NLL score only in the entire paper was written by respected authors in a well renowned ML venue similar to ICLR. We have taken the datasets that have been used by a lot of papers, and benchmarks reported by several academic institutions. Further, we have clearly stated that this paper was written in an academic setup with limited resources and that should be taken into consideration for limited yet convincing experimentation.
> We have added results on 1000 hours of LibriSpeech tokenized and Audio Classfication and they continue to hold true.

---

> > ### Author Response · Authors · 2024-11-28
> > **Response 2**
> >
> > Question 1 We can only say that it yields to better convergence both in terms of speed and the performance. If I can train the same model in almost half the time achieving the same performance, without adding any parameters that is a huge win. We show this happening across modalities and several datasets including two more experiments on acoustic tokens and audio classification.
> >
> > Question 2: Yes, we will include that in the camera ready version as to what characteristics these kernels exhibit and do they generalize across datasets or are they fixed amongst every learning algorithm.
> >
> > Question 3: We have added three results now with learnable and three with non-learnable kernels. The reason is purely computational, and running these models is very expensive to train from scratch. That being said there should be a place in academia for these kind of papers otherwise such approaches will never see the light.
> >
> >
> > Question 4: It would be the same except the filters on 1-D Transformer embeddings being learned. We can include a visual map in the appendix in the camera ready version.
> >
> > Question 5: Yes, we will do it in the camera ready version if accepted.

---

> ### Comment · Reviewer_G6qn · 2024-11-29
>
> Thank you for your reply. Thank you for the clarification on the token length and the wavelet response. What is the limitation of the computational resource you used? Could you please provide some calculation on the difficulty to train a model with 300-500M in your machine?
>
> > The paper currently under review is here, in front of us. What another paper written by reputed authors in another reputed ML conference did or did not is none of my concern: every paper is evaluated within it's own context. What is reasonable within the context of another paper is not always appropriate for another. What the author wrote sounds like an appeal to established authority: that X reputed authors did Y and got accepted to a similar level of conference, so it must be right. That is not grounds for evaluating what this paper has presented.
>
> Personally, I agree with reviewer Tyuw about this. I disagree with some of his arguments on the weakness of your paper, but I suggest a more decent discussion in order to gain more support from the meta-reviewer and area chair.

---

> ### Author Response · Authors · 2024-12-03
> **Response about computational resource**
>
> Dear Reviewer,
>
> The updated paper is here with extra experiments.
> https://drive.google.com/file/d/1u9OGe6VwnKhkyIBs9-k5a87c-rVFF637/view?usp=sharing
>
> | Modality                | Baseline | Proposed | SP Epoch     | SpeedUp | Rel. GPU Hrs |
> |-------------------------|----------|----------|--------------|---------|--------------|
> | Text-8                 | 0.93     | 0.92     | 14.5 epochs  | 42%     | 1.013        |
> | Raw Audio              | 1.84     | 1.70     | 3.7 epochs   | 85%     | 1.042        |
> | Symbolic Music         | 2.08     | 2.02     | 13 epochs    | 48%     | 1.059        |
> | Text-8 (Learnable)     | 0.93     | 0.91     | 12.9 epochs  | 48.4%   | 1.094        |
> | Wiki-103 (Learnable)   | 4.11     | 4.05     | 9.5 epochs   | 62%     | 1.130        |
> | LibriSpeech (Learnable)| 2.43     | 2.40     | 9.2 epochs   | 63.2%   | 1.110        |
> | FSD-50K                | 40.6%    | 42.8%    | 32/92 epochs | 65.2%   | 1.037        |
>
>
> I have made it very clear from the outset that this paper was written in an academic setup with limited computational resources. Training such a model would take weeks on 1-2 GPUs at our disposal. The current models itself take about a day to run. That being said they are exactly following the architecture topology except for being shrunk down in terms of the number of heads, model dimension and layers.
>
> If the experiment of 300M-500M is an absolute necessity, I will run them in the camera ready version.
>
> I agree with your statement that the grounds of evaluating the paper cannot be used in context of another paper, and in no ways was establishing an authority -- rather we give an example of similar paper report the metrics. We have in this paper tried to run it on many modalities including images at pixel level, byte level text, math expressions for LRA and acoustic tokens, characters, GPT-2 tokens, and raw audio samples, in addition to raw waveform patches now and continue to see the result holding true.

---

> ### Comment · Reviewer_G6qn · 2024-12-03
>
> Thank you for the explanation. It looks good to me now.

---

### Official Review · Reviewer_4yst · 2024-11-02

**Soundness:** 2
**Presentation:** 1
**Contribution:** 2
**Rating:** 3
**Confidence:** 3

**Summary:**

This paper introduces WaveletGPT, a method that integrates wavelets with Large Language Models (LLMs). By applying a multi-scale structure to intermediate embeddings in a GPT-style LLM architecture, WaveletGPT achieves pre-training performance nearly twice as fast in text, raw audio, and symbolic music without adding much parameters. The architecture ensures that each token prediction can access intermediate embeddings at varied temporal resolutions in every layer. The effectiveness of WaveletGPT is demonstrated using three open-source datasets: text-8 for natural language, YouTube-Mix-8 for raw audio waveform, and MAESTRO for symbolic music. Additionally, the architecture's performance on the Long Range Arena (LRA) benchmark tasks shows significant improvements across all three modalities.

**Strengths:**

1. The authors proposed an almost free technique to help training LLMs from scratch, without the need of pre-trained larger teachers or significant modifications of the model architecture.
2. The proposed technique is simple but plausible. The authors also tried to address the feasibility in areas like audio and symbolic music.

**Weaknesses:**

1. The idea of leveraging wavelets in Transformers is novel. However, the exploration of wavelet transform is insufficient in the paper, instead, the technique degrades to hand-crafted intermediate convolutional layers between the Transformer blocks (averaging and upsampling is just a special case of a convolutional layer). So extremely speaking, the authors simply added multi-scale convolutional layers between Transformer blocks to speed up convergence, which is rather intuitive. This undermines the innovativeness.
2. Still the concern, the authors claim that the proposed method is superior because it gathers multi-scale information. However, there is no evidence to support that a single-scale averaging/convolution operation is underperformed (i.e., an averaging operation or convolutional layer with fixed kernal length).
3. There is a lack of ablation experiments, and many of the designs are assumed rather than experimentally supported. Details are listed in the Questions section.
4. The experimental setup is unclear, which brings huge obstacles for reproducibility. The authors repeatedly mention "academic setup" without describing the actual setup they have.
5. The writing is poor, with countless gramatical errors and repeating sentences.

**Questions:**

1. line 127-128, the authors claimed that the proposed method can be easily extrapolated to state space architectures. However, there is no evidence or validation in this paper to support this claim.
2. line 250, what is this "$l$" in $xn^l_{(i)}(k)$? Is that the layer index defined before? Because the fonts are different, and according to the definition in paragraph 189-201, this could also stand for wavelet coefficient level.
3. line 248, if in this case i <= E/2, F will be a negative integer, f(i) will be less then 1, is this a mistake?
4. line 433, the authors mentioned a 32-dim version of the proposed model, where the performance was illustrated in Figure 4. However, in line 384, Figure 4 is mentioned to refer the full size model. In addition, in line 450, Figure 4 is mentioned to refer the learnable version. So which architecture does this diagram actually refer to? Is there a contradiction here?
5. line 115-116, the authors used an audio dataset, YouTube-Mix-8, for "long-context modeling". However, the context length remains still, which is 512 and is identical to other datasets. So why did the authors emphasize long context? Furthermore, in common practice of audio language modeling, a codec tokenizer is utilized to encode and compress audio signals to yeild short representations, because, as also mentioned in line 389, a context length of 512 only represents 32ms of audio, which is practically meaningless. Why didn't the authors tokenize the audios first nor expand the context length?
6. line 416, the authors used an $\alpha$ linearly varying between 0 and 1. However, $ 0 < \alpha < 1 $. So, what are the boundary values of $\alpha$ at the two endpoints? Also, in line 417-418, what are the other scores regarding the audio and music?
7. Still considering $\alpha$, why an $\alpha$ linearly varying between 0 and 1 is equivalent to the proposed multi-scale wavelet transform? The smallest kernel size of the proposed transform is 2, so it's the average of 2 tokens, which is approximately equivalent to the situation where $\alpha = 0.5$. The fairness of this comparison needs further verification.
8. There is a lack of ablation experiments, and many of the designs are assumed rather than experimentally supported. For example, why did the authors manipulate exact half of the coordinates of the hidden representations? What will happen if we manipulate more? Less?
9. If I understand correctly, the proposed method is implemented with two feed-forward-networks (FFN), mentioned in line 360-361, while the baseline (a scaled-down version of GPT-2) still consists of one FFN each layer. This introduces siginificant parameter advantage and makes the comparison unfair. Please correct me if I understood wrong.
10. line 448, the authors mentioned that the learnable convolution layers only introduce 0.02M extra parameters. However, I believe a 10-layer Transformer with a hidden dimension of 128 inherently contains fewer parameters, so an absolute number is meaningless. What is the proportion of newly introduced parameters? What is the proportion of newly introduced FLOPs?
11. There are also countless grammatical errors and typos in the paper, such as:

    a. line 48, "Hinton et al. (2015)" should be "(Hinton et al., 2015)"

    b. line 53-79, repeating sentences.

    c. line 127, "Large Language models" -> "Large Language Models" (or just LLM, since mentioned before)

    Most of the grammatical errors are too minor to list individually here. I know I shouldn't be overly demanding about writing standards, as the idea is what matters most, but the writing errors in this paper is so much that I can't help but question the author's rigor.

**Details Of Ethics Concerns:**

No concern.

---

> ### Author Response · Authors · 2024-11-28
> **Response 1 to Weakness**
>
> We have added several more experiments now, particularly for acoustic tokens and Audio Classification.
>
> The updated paper is here with extra experiments. https://drive.google.com/file/d/1u9OGe6VwnKhkyIBs9-k5a87c-rVFF637/view?usp=sharing
>
> Weakness 1 I beg to differ on this. In Figure 2, you will see convolutional and pooling layers. Does that mean we call wavelets a convolutional neural network? NO. The title itself says that it is inspired by a wavelet filter bank. One also cannot compute wavelet transform for the entire sequence length as it requires the whole length to be computed as the transform of, and that would undermine the causality assumption. In order to mitigate these constraints, we added multi-scale convolutional filters across embedding dimensions and showed that one can still draw connections with Haar wavelet with varying window lengths. I would disagree with undermining the innovativeness as there has been no prior work that could put a constraint on multi-scale structure embedding in every layer, preserving the causality assumption while increasing the convergence speed and performance with addition to no extra parameter or < 0.2% of them. Further, the results continue to hold true on a wide variety of datasets.
>
>
> Weakness 2: We have done an average pooling operation to compare with EMA and found that the performance degrades when we carry out EMA. The hyperparameter to explore what kernel length should be optimized would be far more than the compute resources available to us. Further, doing a single scale averaging convolution would no longer be akin to a wavelet tree structure. There has been enough evidence in speech and signal processing that multi-scale convolutions perform better than single-scale convolutions in applications such as ASR. The whole body of wavelet literature and the books that have been written emphasize multi-scale structure computation. If the paper is accepted, we will include experiments related to it.
>
>
> Weakness 3 This paper requires substantial computation, and it is not possible to train everything from scratch so many times. I would also like to make a point here: We need not populate our paper with details and experiments that are not necessary, and doing a lot of ablation studies should not be a criterion for acceptance. We have reported the results on four different modalities with the new addition of acoustic tokens, and in addition to that, we have reported on audio classification benchmarks. Further, we have reported on LRA benchmarks on three datasets and done a study on the effect of context length and the number of layers that were the most important. The paper has been tested on raw pixels, raw audio, byte text representation, GPT-2 tokens, characters, acoustic tokens, and math expressions, which shows the strength of our work.
>
>
> Weakness 4 We have described our model in as much detail as one could in the experimental setup. It is a shrunk-down version of GPT for doing pretraining and a variant of it for LRA benchmarks. Everything is mentioned to have the best possible reproducibility, including the architecture, learning rate schedules, number of epochs, and the number of layers. We have explained our algorithm as clearly as we could. In addition, we will also open source our codebase if the paper is accepted.
>
> Weakness 5 This is for sure not an English writing conference. We have improved the grammar and ran through spell checks, and there are almost no errors present to the best of our ability. We have removed a single repeated sentence that was present.

---

> ### Author Response · Authors · 2024-11-28
> **Correct Response to Questions**
>
> We have improved the responses.
>
> Q1 We do not need to. The method causally modifies the intermediate embedding. We have shown that our method works on embedding space, and the architecture should not matter.
>
> Q2. We are not sure about this point. Yes, it could stand for wavelet coefficient level, but it does not.
>
> Q3 Nope, this will never happen, and f(i) will always be greater than or equal to 1. There is a typo in the paper and the less than or equal to should be greater than or equal to and vice-versa. Thank you for catching this error.
>
> Q4 Nope, why is this confusing? All of the algorithms can be easily tweaked for model dimension, whatever we want. All of the models in Table 1 are full-size models. We see how the model does for the number of layers and model dimension reduced. We see the same effect, i.e. faster convergence by 40-60% and better performance, which makes the work exciting as the performance trend can be seen for increased layer and model dimension. The algorithm should work for any model dimension. We have clearly stated it in a separate section for LRA benchmarks.
>
>
> Q5 This is because it is a benchmarked dataset used for long-context modelling. We have also added experiments on LibriSpeech on ENCODEC tokens, which contain 1000 hours of speech in the setup and are typically used to model coarse acoustic tokens. We wanted to add much diversity in showing that the method works for audio waveforms, audio tokenizers, images, math expressions, text represented by bytes, letters, and GPT-2 tokenizers. Expanding context length adds more computing time for all of the answers. We have shown through LRA benchmarks, which tested model ability in the long context of 1999, 3172, and 2048, that the method continues to hold. There is no change to our algorithm or architecture at all except handing different context length and model dimensions which is straightforward.
>
> Q6 Yes, the numbers vary linearly, with 0 and 1 excluded. We just did it for one modality and need only do it for some audio and music datasets. Transformers are ubiquitous in modelling tokens as long as some dependencies exist across tokens. So, we do not need to do every experiment for every modality.
>
> Q7 I will add multi-resolution EMA as a standard method for both the points mentioned above, 6 and 7. We implement a standard averaging method to compare the performance of our method with fairness, similar to how a multi-resolution EMA is carried out with our proposed method. It is fair as EMA does not have a concept of kernel size.
>
>
> Q8 We agree that we should have done more ablation experiments on how many coordinates to manipulate. Does it help to over-constrain Transformer Decoder embeddings or under-constrained them? Is half the sweet spot? We will run an experiment for the camera-ready version, at least for one of the modalities, on the performance of pre-training vs the ratio of embeddings manipulated. Additionally, let us assume that half was not the sweet spot, and perhaps we needed to manipulate only a quarter of the embedding coordinates. This will only improve the results, and we have already reported quite strong results regarding convergence speeds and performance boosts.
>
>
> Q9 Instead of a single-layer MLP, we use a two-layer MLP in both. We would not do that and make the comparison unfair. For both the baseline and modified GPT, we use a two-layer MLP instead of a single-layer MLP inside a Transformer Decoder block. I hope it makes it clear now.
>
>
> Q10 We have added the clarification in the paper where it is <0.2% parameters for around 10 million parameter architecture. We compare against GPU hours, i.e., the total time taken to train the baseline architecture and the ratio of the time the new architecture takes to train. As we can see, it adds a minuscule relative training time with a similar negligible increase in flops.
>
>
> Q11 We have corrected the errors mentioned above. Most errors have been corrected with spell checks and other automated grammar tools.

---

> ### Author Response · Authors · 2024-12-03
> **Extra Results**
>
> Here are more experiments added for 1000 hours of LibriSpeech and Audio Classification on FSD-50K
>
> | Modality                | Baseline | Proposed | SP Epoch     | SpeedUp | Rel. GPU Hrs |
> |-------------------------|----------|----------|--------------|---------|--------------|
> | Text-8                 | 0.93     | 0.92     | 14.5 epochs  | 42%     | 1.013        |
> | Raw Audio              | 1.84     | 1.70     | 3.7 epochs   | 85%     | 1.042        |
> | Symbolic Music         | 2.08     | 2.02     | 13 epochs    | 48%     | 1.059        |
> | Text-8 (Learnable)     | 0.93     | 0.91     | 12.9 epochs  | 48.4%   | 1.094        |
> | Wiki-103 (Learnable)   | 4.11     | 4.05     | 9.5 epochs   | 62%     | 1.130        |
> | LibriSpeech (Learnable)| 2.43     | 2.40     | 9.2 epochs   | 63.2%   | 1.110        |
> | FSD-50K                | 40.6%    | 42.8%    | 32/92 epochs | 65.2%   | 1.037        |

---

> ### Comment · Reviewer_4yst · 2024-12-03
> **Official Comment by Reviewer 4yst**
>
> The authors' rebuttal responses are as difficult to read as the paper itself. I raised 11 questions in the review process, but the authors provided 12 answers, some of which I am unsure which question they correspond to (For example, is the answer to "Q4" really about question 4? I will tentatively take it as the answer to question 3.)
>
> For most of the questions I raised, the author did not truly provide explanations. For example, the authors use "this will never happen" to answer question 3, which is confusing. If i <= E/2, then F = int(L_k * (i - E/2)/(E/2 - 1)) is clearly negative or zero. I don't think "this will never happen" is an answer. Also, as for question 6, when I was asking about the boundary values of $\alpha$, I was asking about the actual boundaries, like 0.05? 0.95?
>
> **My main concern** is that the authors use the argument "there is no need to conduct unnecessary ablation studies" to sidestep questions about their choices of key designs. I agree that unnecessary experiments should not be conducted, but what defines unnecessary? The authors did not provide any explanation and simply classified my doubts as unnecessary. If the experiments are really impossible to conduct, the authors could just provide the reasons or discussions for the current parameter setting. I believe that merely arguing verbally is ineffective and can negatively impact the academic environment at ICLR. Because I could very well submit an LLM-related paper lacking ablation experiments, noting in the paper that "we are in an academic setting so we only train a 10M model and do not conduct experiments with larger-scale models or parameter verification," nor discuss scalability-related issues, and then during the rebuttal phase, I could emphasize "we will not conduct so many additional experiments in such a short rebuttal period." I think such practices are contrary to the spirit of ICLR.

---

> > ### Author Response · Authors · 2024-12-03
> > **Corrected Typo in Algorithm + Ablation Concern +   Correct Order**
> >
> > ** We have corrected the comment order. We apologize for the confusion before.
> >
> > ** There was a typo present in the Algorithm 1. All of the less than and equal to sign should be great than and vice versa. We are sorry for the slip-up; it is easy to overlook. It has been corrected, and we have added it here as well.
> >
> > **Algorithm 1: Wavelet-GPT**
> >
> > 1. **Input:**
> >    - $E$: Model or Embedding Dimension
> >    - $L$: Context Length
> >    - $N+1$: Number of Decoder Layers
> > 2. **For** layer $l = 1, 2, ..., N$:
> >    1. $\mathbf{x}^l \gets$ Output of Transformer $l^{th}$ Decoder Block // Dimension $E$ x $L$
> >    2. $\mathbf{xn}^l \gets$ Modified Transformer Embedding replacing $\mathbf{x}^l$
> >    3. **For** Embedding dimension $i < E/2$:
> >       - $\mathbf{xn}^l_{(i)} \gets \mathbf{x}^l_{(i)}$
> >    4. **For** Embedding dimension $i \geq E/2$:
> >       - $\mathbf{f(i)} \gets 2^F$ where $F = int(L_k*(i-E/2)/(E/2-1))$ and $L_k = \lfloor \log_2(L) \rfloor + 1$
> >       - $\mathbf{xn_{(i)}^l(k)} \gets \frac{1}{f(i)} \sum_{m=k-f(i)}^k x_{(i)}^l(m)$ // For Non-learnable fixed Haar wavelet
> >       - $\mathbf{xn_{(i)}^l(k)} \gets \sum_{m=0}^{f(i)-1} h(m) \cdot x_{(i)}^l(k-m)$ // For learnable wavelet kernel $h$
> >
> > ** EMA: We vary alpha linearly from (0,1), excluding the endpoints. The alpha step size will vary depending on the value of the embedding dimension since we only carry out half of the coordinate dimension. For 32, the step size will be large, and for 128, it will be small and linearly spaced.
> >
> >
> > ** Ablation Concern:
> >
> > Our Response: We have tried to explain politely and respectfully and are not arguing.  Comments like "can negatively impact the academic environment at ICLR. " are just not right. Please choose your words carefully in public comments. We are not side-stepping the key design choices. Nor are we negatively impacting the academic environment.
> >  You have a choice to give whatever score you deem fit for this paper.
> >
> > We have addressed the fact that the paper was already written on limited academic computing resources. I do not want to publicly mention the reasons for limitations on computing, and I hope they are respected.
> >
> >
> > ** Response to this ==
> > ""Because I could very well submit an LLM-related paper lacking ablation experiments, noting in the paper that "we are in an academic setting, so we only train a 10M model and do not conduct experiments with larger-scale models or parameter verification,"
> >
> > Our Response: There is nothing wrong with this in an academic setup. Someone's larger model is another institution/company's tiny/smaller model, and it is all relative. So please do not be disrespectful.
> >
> > ** emphasize "We will not conduct so many additional experiments in such a short rebuttal period." I think such practices are contrary to the spirit of ICLR.
> >
> > Our Response: We have the absolute right to choose which experiments we want to conduct and in what manner during the rebuttal phase. There is nothing wrong with this. It should never be the spirit of ICLR to try to cram as many experiments in a short span of 2 weeks from four reviewers.
> >
> > ** Ablation Study:
> > Our Response: We have tested to make the model smaller, reducing the number of layers, and the result continues to hold. Further, we tested varying context lengths of 1999, 3172, and 2048 in addition to our context of 512. We have not scaled our model in the 100-500M regime and would love to do so in the camera-ready version of the paper in the next month or so.
> >
> > We have not conducted an ablation study to modify the ratio of the manipulated embedding coordinates. For the argument's sake, let us assume that only a quarter or 3/4th of the coordinates had to be modified as the optimal choice instead of half. In that case, the paper results will become even stronger than the currently reported ones as they will yield a better convergence/performance boost. If not, the current results are already promising.
> >
> > Thank you for your time in giving your suggestions and being critical.

---

### Official Review · Reviewer_HDRD · 2024-11-03

**Soundness:** 3
**Presentation:** 2
**Contribution:** 3
**Rating:** 6
**Confidence:** 3

**Summary:**

The authors propose a method to encode multi-resolution context in GPT-style LLMs inspired by haar wavelet transform.

The method works by modifying the intermediate embeddings of the transformer blocks: replacing half the embeddings by the average pooling (of this half) over the context with windows growing exponentially in size.

**Strengths:**

The proposed method is simple with minimal computation overhead but results in performance improvements and a great reduction in the training time. The authors show this in different modalities and datasets. I feel the idea of encoding the context as an inductive bias is relevant for the community and worth studying.

**Weaknesses:**

- I think the study has been done in a small setup, architecture, context length, and datasets; it's hard to tell if these findings would scale up, as the baseline does. From the results, it seems this method helps by sacrificing half the embedding space to average pooling over the context (albeit with different windows); it's hard to tell if this would be the case when scaling to more difficult tasks and longer context.
- Table 1, lacks the complete results for learnable and unlearnable kernels on all the datasets.
- Although the authors motivate their method by wavelet transform from signal processing, I think the final method can be simpler to describe, as it's implemented as simply an average pooling operation (with different window sizes; window size is a function of the band).

Minor Issues:


- Lines 53 and 76, the text is duplicated.
- Line 114 typos
- Sections 4.3, 4.4 and 4.5, I think it's better to summarize the results in a table.

**Questions:**

In addition to the weaknesses above, I have the following questions:


- How do you deal with residual connections for the modified embedings 0:E÷2 . I mean, in the top transformer blocks, these embedings have residual connections coming from the modified (pooled) embedings. Is this the optimal setup?

- Why choose the avg-pooling on the embedings instead of biasing the attention mechanism like WavSPA?

- From Line 247, it seems the max window size is determined during training to be the context L, will this hinder the ability to expand the model context size after training ?

---

> ### Author Response · Authors · 2024-11-28
> **Response 1 to Weaknesses**
>
> We have added several more experiments to include audio classification, acoustic token based experiments on 1000 Hours of LibriSpeech
>
> The updated paper is here with extra experiments. https://drive.google.com/file/d/1u9OGe6VwnKhkyIBs9-k5a87c-rVFF637/view?usp=sharing
>
> We want to thank the reviewer for giving us a good score. We agree that the idea of encoding the context as an inductive bias and the broader theme of integrating signal processing algorithms with that of LLMs is worth exploring.
> Our Response: We agree that the study has been done in a small setup only for the number of parameters. The context length for long range arena benchmarks are quite long ranging from 3072 for images, 2048 for byte encoded text and 1999 for ListOps as is the standard practice. We are not sacrificing half of the embedding space == they will learn representations that capture the hierarchical structure present in the data. So it will learn embeddings that can capture 2/4/8/16 tokens as it deems fit. So in some of the embedding dimensions it will just have one coordinate descriptive of the entire sequence whether it is the topic model or for music genre or composer or for speech the speaker ID and so on. This is also motivated in our introduction that data around us is inherently hierarchical with a structure.
>
> Table 1, lacks the complete results for learnable and unlearnable kernels on all the datasets.
> Our response: This has been deliberate and does not lack the results. We all can agree that the learnable kernel optimized for next token prediction should give better performance as opposed to fixed non-learnable kernel. We have added more experiments on speech acoustic tokens for 1000 hours of speech, scaling Wiki103, audio classification and machine translation. These experiments themselves are quite GPU intensive and as we have emphasised repeatedly that these experiment was carried out in an academic setting with resource constraint setup in terms of compute available. We have push the experiments to more than the limit that we could.
> Although the authors motivate their method by wavelet transform from signal processing, I think the final method can be simpler to describe, as it's implemented as simply an average pooling operation (with different window sizes; window size is a function of the band).
> Our response: We would politely disagree. This would be the same as telling that a tree wavelet structure as shown in Figure 1 and Figure 2 with the reference added would be a convolutional network just because it uses convolution and pooling operation. Yes, we can describe learnable and non-learnable operations with pooling and convolution operations but  it would be unfair to simply call it average pooling operation. Further in the learnable section, we learn multi-resolution convolution kernels akin to a wavelet tree structure shown in Figure 2.
>
>
> Questions:
> In addition to the weaknesses above, I have the following questions:
> How do you deal with residual connections for the modified embedings 0:E÷2 . I mean, in the top transformer blocks, these embedings have residual connections coming from the modified (pooled) embedings. Is this the optimal setup?
> There are no residual connections for modified embeddings. As we have put up in the algorithm, simple Transformer baseline would be x1= Transformer(x), x2 = Transformer(x) …. We would just do our embedding manipulations on x1. A Traditional transformer based  GPT or any architecture does not contain any skip connections between successive transformer blocks. It only has skip connections inside the Transformer block.
> Why choose the avg-pooling on the embedings instead of biasing the attention mechanism like WavSPA?
> WavSPA biases attention. Our goal was to mimic real world setup while keeping the attention mechanism free to do whatever it wants to do. The embeddings capture various features of the data, and if we are to say that we are trying to capture hierarchical structure present in various modalities such as images, text, audio, we say that it is present in the embeddings that we derive from these modalities: This was our thinking behind it.
> From Line 247, it seems the max window size is determined during training to be the context L, will this hinder the ability to expand the model context size after training ?
> Yes, the max window size is determined during training to be the context L and will hinder the ability to expand the model context size after training. However to a certain extent the same thing can be said for the transformer model with the same context length, even though we can still operate it on longer sequences the query, key and value weights are learned only for the context length L

---

> ### Author Response · Authors · 2024-12-03
> **Extra Results on 1000 Hours of LibriSpeech and FSD-50K**
>
> | Modality                | Baseline | Proposed | SP Epoch     | SpeedUp | Rel. GPU Hrs |
> |-------------------------|----------|----------|--------------|---------|--------------|
> | Text-8                 | 0.93     | 0.92     | 14.5 epochs  | 42%     | 1.013        |
> | Raw Audio              | 1.84     | 1.70     | 3.7 epochs   | 85%     | 1.042        |
> | Symbolic Music         | 2.08     | 2.02     | 13 epochs    | 48%     | 1.059        |
> | Text-8 (Learnable)     | 0.93     | 0.91     | 12.9 epochs  | 48.4%   | 1.094        |
> | Wiki-103 (Learnable)   | 4.11     | 4.05     | 9.5 epochs   | 62%     | 1.130        |
> | LibriSpeech (Learnable)| 2.43     | 2.40     | 9.2 epochs   | 63.2%   | 1.110        |
> | FSD-50K                | 40.6%    | 42.8%    | 32/92 epochs | 65.2%   | 1.037        |

---

### Meta-Review · Area_Chair_D86B · 2024-12-18

**Metareview:**

The paper proposes the use of wavlet functions in Transformers.

I recommend rejection because of two reasons: 1) poor writing and 2) poor experimental methodology.

Reviewers HDRD, 4yst, and Tyuw raised the writing issue. It is also pretty clear from the abstract and the introduction that the paper requires a significant revision. In fact, the introduction is one giant paragraph. Besides the motivation, the presentation of wavelet can be significantly simplified and shortened, as pointed out by reviewers.

Reviewers G6qn, 4yst, and Tyuw were concerned about the experimental methodology. It's best to follow a well accepted pre-training and evaluation protocol. In addition, the paper should conduct controlled experiments and report ablation results.

**Additional Comments On Reviewer Discussion:**

The discussion had mostly around clarification questions, a sign that the paper was not well written enough that the key facts can be easily retrieved. The authors had provided experiments as evidence, and those did address some of the concerns raised by the reviewers.

---

### Decision · Program_Chairs · 2025-01-22

Reject